# An Efficient Virtual Machine Consolidation Algorithm for Cloud Computing

**DOI:** 10.3390/e25020351

**Published:** 2023-02-14

**Authors:** Ling Yuan, Zhenjiang Wang, Ping Sun, Yinzhen Wei

**Affiliations:** 1Department of Computer Science, Huazhong University of Science and Technology, Wuhan 430074, China; 2School of Information, Wuhan Vocational College of Software and Engineering, Wuhan 430074, China; 3School of Computer Science, Huanggang Normal University, Wuhan 430074, China; 4School of Computer Science, Wuhan Vocational College of Software and Engineering, Wuhan 430074, China

**Keywords:** virtual machine consolidation model, load prediction, virtual machine migration, blockchain

## Abstract

With the rapid development of integration in blockchain and IoT, virtual machine consolidation (VMC) has become a heated topic because it can effectively improve the energy efficiency and service quality of cloud computing in the blockchain. The current VMC algorithm is not effective enough because it does not regard the load of the virtual machine (VM) as an analyzed time series. Therefore, we proposed a VMC algorithm based on load forecast to improve efficiency. First, we proposed a migration VM selection strategy based on load increment prediction called LIP. Combined with the current load and load increment, this strategy can effectively improve the accuracy of selecting VM from the overloaded physical machines (PMs). Then, we proposed a VM migration point selection strategy based on the load sequence prediction called SIR. We merged VMs with complementary load series into the same PM, effectively improving the stability of the PM load, thereby reducing the service level agreement violation (SLAV) and the number of VM migrations due to the resource competition of the PM. Finally, we proposed a better virtual machine consolidation (VMC) algorithm based on the load prediction of LIP and SIR. The experimental results show that our VMC algorithm can effectively improve energy efficiency.

## 1. Introduction

Most of the data collected by IoT devices on the blockchain system are processed in the cloud. The large amount and high speed of real-time data sent on IoT devices pose a severe challenge to cloud computing methods [1]. However, due to the dynamic changes of the application load on the virtual machines (VMs), the heterogeneity of physical machine (PM) resources in the cloud data center is often out of balance. Some PMs are overloaded, which may result in resource competition and service level agreement violation (SLAV) [2]. Some PMs are underloaded, which may result in low resource utilization and reduce the energy efficiency of the data center [3]. There is an urgent need for algorithms that can effectively improve the energy efficiency and quality of service of cloud computing in the blockchain.

The VM dynamic migration technology can dynamically change the host PM of the VM at runtime [4]. Virtual machine consolidation (VMC) technology means that the VM manager periodically and dynamically deploys VMs to minimize the number of running PMs and simultaneously turn off the idle PMs according to the load conditions of each PM [5]. VMC can effectively reduce the imbalance of the data load in the blockchain and save energy consumption. For an overloaded PM, some of the VMs in it can be moved out to avoid further degradation of performance to reduce the SLAV rate; for an underloaded PM, all VMs on it can be migrated out, and it can be shut down to minimize the number of PMs to save energy [6,7].

However, the dynamic changes in the VM load make it extremely difficult to design an efficient VMC algorithm. Two problems to be solved by VMC in blockchain are as follows [8]:(1)For an overloaded PM, which VMs on it need to be migrated out;(2)For a VM to be migrated out, which PM is the better migration destination.

This paper focuses on the above problems, and our contribution is as follows:(1)We proposed a migration VM selection strategy called LIP based on load increment prediction to improve selection accuracy. It quantitatively analyzes the growth trend of the VM load and combines the current load with load increment. We designed a volatility-based weighted time regression prediction algorithm called VWTRP to improve the accuracy of load trend prediction.(2)We designed a VM migration point selection strategy called SIR based on load sequence prediction and optimal saturation increase rate. SIR can improve the stability of PM load in data centers by leveraging the complementary effects of load sequences between VMs to be migrated and VM migration points. We proposed a load prediction algorithm called LSMP based on load similarity matching to compensate for the shortcomings of load stationarity matching, which only depends on historical load sequences.(3)Combining LIP and SIR, we designed a VMC algorithm based on load forecasting. We used actual load data for simulation experiments and evaluated the proposed algorithm. The experimental results show that our proposed algorithm improves performance in accuracy and stability.

The rest of this paper is structured as follows. Section 2 shows the related work of virtual machine consolidation technology. Section 3 introduces our proposed algorithm, which includes short-term load increment prediction and load sequence prediction, our migration VM selection strategy LIP, the VMs migration point selection algorithm SIR, and the consolidation algorithm that combines them. Section 4 introduces the experimental methods and analyzes the experimental results. Section 5 is the conclusion.

## 2. Related Work

In terms of the VM migration selection strategy, Verma et al. [9] designed three kinds of selection strategies: random choice (RC), highest potential growth (HPG), and minimum migration time (MMT). Beloglazov et al. [10] proposed three kinds of selection strategies, viz., minimum migration time (MMT), random selection (RS), and maximum correlation (MC). The minimization of VM migration (MVM) algorithm was proposed by [11]. Masoumzadeh et al. [12] proposed an adaptive threshold-based algorithm using dynamic fuzzy Q-learning (DFQL) method. The proposed algorithm learns to consider a VM as an overloaded host according to its energy–performance trade-off optimization. In recent years, the dynamic detection of the load on PMs has drawn scholars’ attention. Bui et al. [13] deployed regression models on the historical utilization of PMs to enhance detection accuracy. Hummaida et al. [14] proposed a reinforcement learning management policy. The method can run decentralized and achieve fast convergence toward efficient resource allocation, resulting in lower SLA violations.

The current methods have the following problems. Current scholars mainly focus on predicting the load on PMs to improve the efficiency of VMC. For the selection of the migration VM, only the current resource load of the VM is focused, and the dynamic characteristics of it are not analyzed. Although some methods using reinforcement learning have been proposed, most of them are time-consuming and energy-consuming, and are not particularly suitable for industrial scenarios. In addition, the current strategies of VM selection only restore the PM load to a normal level and neglect the root cause of the load growth. The improper migration of VMs causes an increase in SLAV and the number of migrations.

In terms of VM migration destination selection strategy, Dabbagh et al. [15] proposed random choice (RC) and first fit (FF). Pascual et al. [16] proposed next fit (NF)/round robin (RR). Farahnakian et al. [17] proposed best fit (BF). Khaleel et al. [18] calculated the deviation between the real utilization of the running server and its threshold, then picked the VM whose VM utilization is close to the deviation. Chen et al. [19] proposed an algorithm using host utilization and minimal correlation methods to predict the future utilization of the VMs and then use the predicted data to select the migration destination. Rjoub et al. [20] applied machine learning methods and used statistics of CPU load to predict the most proper destination.

The current methods have the following problems. When selecting a VM migration point, they also only focus on the current resource load of the VM, but fail to consider the VM resource load as a time series, ignoring the complementary effects between the VM load sequences. If the complementary effects of the resource load sequences of each VM in the PM are poor, the load fluctuation of the PM will be enhanced, which means frequent VM migration. Although some methods used the strategy of machine learning, they just focus on statistical data and need deeper data mining.

We proposed a better virtual machine consolidation (VMC) algorithm based on the load prediction which is the organic combination of LIP and SIR. The experimental results demonstrate that the proposed VMC algorithm can outperform the existing schemes in terms of efficiency.

## 3. Algorithm Description

In this section, we proposed 7 algorithms to solve the problem of virtual machine consolidation. To help readers quickly understand the content of this section, we provide two tables which organize the abbreviations of the algorithms and notations of every parameter. The table of abbreviations of proposed algorithmsis given in Table 1 and the table of notations is given in Table 2.

### 3.1. Short-Term Load Increment Prediction

In order to ease the growth trend of the load in overload PM from the root, the VM whose load has a growing trend should have a high priority to be migrated out. Therefore, we proposed an algorithm, namely volatility-based weighted time regressive prediction (VWTRP), for short-term load trend prediction for VMs, which is the basis of the selection strategy for VMs to be migrated out.

Program flow chart of the algorithm is shown in Figure 1.

The VWTRP algorithm divides the VMs set into three VMs sets with different priorities according to the load increment.

The VWTRP algorithm is shown in Algorithm 1. The time complexity of it is $O(n^{log7})$, where *n* is the number of acquisition periods in the consolidation period. The VWTRP algorithm is composed of the following four important steps.
**Algorithm 1: **VWTRP(*v*, LH)**Input:** 
VM *v*, Historical load data of *v*
LH**Output:** Load increment LI1:Calculate load sequence ls[n+1]2:Calculate reverse orders number *A*3:Calculate *Z*4:**if**Z<=1.96**then**5:     **for** i=1 to n−1 **do**6:         Calculate volatility v[i]7:         Calculate weight w[i]8:     **end for**9:     curve(t)←timeRegression(ls[],w[])10:   Calculate $R^{w}$11:   Calculate LI12:    **if** LI≤ls[n]∗H∗T **then**13:         $V_{m}\leftarrow V_{m}\cup v$14:    **else**15:         $V_{h}\leftarrow V_{h}\cup v$16:    **end if**17:**else**18:    $V_{l}\leftarrow V_{l}\cup v$19:**end if**20:**return**LI

#### 3.1.1. Inspection of Load Growth Trends

The load growth trend was examined based on the number of reverse order. For the time series $y_{1}y_{2}...y_{m},if\;y_{i}<y_{j}$, then it is called a reverse order, where i,j∈{1,2,...,M} and i<j. The total number of reverse orders is denoted by *A*. The expectation and variance with respect to *A* are shown in (1) and (2) [21].
(1)$$\begin{array}{c} E\left(A\right)=\frac{1}{4}M\left(M-1\right) \end{array}$$
(2)$$\begin{array}{c} D\left(A\right)=\frac{M\left(2M^{2}+3M-5\right)}{72} \end{array}$$
where *M* denotes the number of data in time series. Then, according to the above Equations, the statistic *Z* which is given in (3) is a standard normal random variable, that is Z∼N(0,1):(3)$$Z=\frac{\left[A+\frac{1}{2}-E\left(A\right)\right]}{\sqrt{D(A)}}$$

In the case of significant level α=0.05, if Z>1.96, then it has a growing trend.

#### 3.1.2. Load Volatility and Weight Calculation

In the acquisition period $\left[t,t+T_{s}\right]$, suppose that the acquisition module receives additional *k* load data items, there should be altogether k+2 load data items, i.e., $u_{0}^{\prime },u_{1}^{\prime },\ldots ,u_{k}^{\prime },$$u_{k+1}^{\prime }$, where $T_{s}$ is the acquisition period.

If $u_{i}^{\prime }>u_{i-1}^{\prime },\;\mathrm{and}\;0<i\leq k+1$, it is marked as “+”; otherwise, it is marked as “−”. A continuous sequence with same symbols is defined as a Run. For example, there are five Runs in the following symbol sequence: +−−−+++−−+. They are +,−−−,+++,−− and +, respectively.

Supposing that the CPU historical load data at the acquisition points $t_{0},t_{1},\ldots ,t_{n}$ in a consolidation period on the VM are denoted by $u_{0},u_{1},\ldots ,u_{n}$, then there are $k_{i}+2$ load data items in $\left[t_{i},t_{i+1}\right]$, and the number of Runs should be $p_{i}$. The load volatility at $t_{i}$ is given in (4).
(4)$$v_{i}=\frac{p_{i-1}-1}{k_{i-1}+2}\times \frac{p_{i}-1}{k_{i}+2}$$

When the load volatility is small, the load data around sampling point do not change significantly and therefore can represent the characteristics of the load. Then, the curve fitting should have a great weight [22]. Hence, the weight of the load data at point $t_{i}$ is shown in (5):(5)$$w_{i}=1-\sqrt{v_{i}},0\leq i\leq n$$

#### 3.1.3. Weighted Curve Fitting and Evaluations on the Curve

We use regression analysis to model the load sequence over time. To avoid overfitting, which affects the accuracy of prediction, the one-variable second-order linear regression model is used in this paper. The one-variable second-order linear regression model for CPU utilization is shown in (6):(6)$$U\left(t\right)=\beta_{0}+\beta_{1}t+\beta_{2}t^{2}+\varepsilon$$
where $\beta_{0}$ is a regression constant, $\beta_{1},\beta_{2},\ldots ,\beta_{p}$ are regression coefficients, and $\varepsilon ∼N\left(0,\sigma^{2}\right)$ is a random error.

The CPU utilization of VM during a consolidation period on the n+1 sample points is denoted by $<U\left(t_{1}\right),\ldots .,U\left(t_{2}\right),U\left(t_{m}\right)>$. The error of the CPU load data on the sample point $t_{i}$ is $\varepsilon_{i}∼N\left(0,{\sigma_{i}}^{2}\right)$.

In the regression analysis, ordinary least squares (OLS) is one of the popular methods used for parameter estimation. The sum of squared deviations using OLS is shown in (7):(7)$$D=\sum_{i=0}^{n}{\left(U\left(t_{i}\right)-\beta_{0}-\beta_{1}t_{i}-\beta_{2}t_{i}^{2}\right)}^{2}$$

However, under the heteroskedasticity, the regression curve will bias those sample points with larger variance of error. When estimating the parameters, the weight calculated in (5) is added in each sampling point. The sum of the squared deviations of the weighted least squares estimate is shown in (8):(8)$$D_{w}=\sum_{i=0}^{n}w_{i}{\left(U\left(t_{i}\right)-\beta_{0}-\beta_{1}t_{i}-\beta_{2}t_{i}^{2}\right)}^{2}$$

The parameter estimated value calculated by wright least squares (WLS) [23] is shown in (9).
(9)$$\begin{array}{c} {\hat{\beta }}_{WLS}={\left(X^{T}W^{T}WX\right)}^{-1}X^{T}W^{T}WU\\ \;\mathrm{where}\;W=\left[\begin{array}{ccccc} w_{0} & 0 & \cdots & 0 & 0\\ 0 & w_{1} & \cdots & 0 & 0\\ \vdots & \vdots & \ddots & \vdots & \vdots \\ 0 & 0 & \cdots & w_{n-1} & 0\\ 0 & 0 & \cdots & 0 & w_{n} \end{array}\right] \end{array}$$

The goodness of fit of the linear regression equation is measured by using a multiple correlation coefficient *R*. However, all the data are treated equally in the process of calculating the multiple correlation coefficient. By analyzing the volatility, the importance of different load data in the time regression analysis is different. Therefore, we should take weights into account when performing model checking. Therefore, we use a weighted multiple correlation coefficient to evaluate the goodness of fit of the regression equation, as shown in (10)–(13):(10)$$R^{w}=\sqrt{\frac{S_{R}^{w}}{S_{T}^{w}}}=\sqrt{1-\frac{S_{E}^{w}}{S_{T}^{w}}}$$
(11)$$S_{T}^{w}=\sum_{i=1}^{m}w_{i}{\left(y_{i}-\bar{y}\right)}^{2}$$
(12)$$S_{E}^{w}=\sum_{i=1}^{m}w_{i}{\left(y_{i}-{\hat{y}}_{i}\right)}^{2}$$
(13)$$S_{R}^{w}=\sum_{i=1}^{m}w_{i}{\left({\hat{y}}_{i}-{\bar{y}}_{i}\right)}^{2}$$
where $R^{w}\in \left[0,1\right]$, the bigger the $R^{w}$, the better the fitting.

#### 3.1.4. Load Volatility and Weight Calculation

The regression function $\hat{U}\left(t\right)$ obtained from the time regression analysis has good performance for short-term predictions. However, it would lead to large deviations for long-term predictions. Therefore, we use the fitting function to predict load at time Δt, then expand it to obtain the load of the next consolidation cycle. At point $t_{i}$, the CPU load increments (LI) in the next consolidation cycle of the schedule are shown by (14)–(16):(14)$$LI=\left(\frac{I}{\Delta t}-U\left(t_{i}\right)\right)T$$
(15)$$I=\int_{t_{i}}^{t_{i}+\Delta t}\hat{U}\left(t\right)dt$$
(16)$$\Delta t=k\times R^{w}\times T$$
where *T* represents the consolidation cycle (there are multiple acquisition periods $T_{s}$ in a consolidation cycle), and *k* denotes the trend continuation coefficient. The larger the *k*, the longer the prediction interval, and the smaller the prediction precision.

### 3.2. The Load Sequence Prediction

There is a strong similarity between the CPU load sequences of each day of the VM. Therefore, the CPU load sequence of the VM in a certain period in the future can be approximated by using the CPU load of a certain past time.

We propose a load similarity match predicted (LSMP) algorithm based on the similarity measurement algorithm to predict the load order of VM. The LSMP algorithm is composed of the following three important steps.

#### 3.2.1. Standardization of Load Sequence

The load sequences are standardized in order to eliminate the effects of amplitude scaling and amplitude translation on the load similarity measure. For the load sequence $X=<x_{1},x_{2},\cdots ,x_{n}>$, the mean value of the set is μ(x) and the variance is $\sigma^{2}\left(x\right)$. The time series can be standardized as $SX=<sx_{1},sx_{2},\cdots ,sx_{n}>$ as shown in (17):(17)$$sx_{i}\left(x\right)=\frac{x_{i}-\mu \left(x\right)}{\sigma (x)}$$

#### 3.2.2. Function Conversion of Load Sequence

The general method based on the comparison of numerical features for the similarity measure of time series ignores the morphological characteristics. To describe the load change information more accurately, we use the piecewise cubic spline interpolation method to convert a load sequence with length *n*
$x_{1},x_{2},\ldots ,x_{n}$ into n−1 smooth cubic curves $f_{1},f_{2},\ldots ,f_{n-1}$.

We use the following method to reduce the complexity of the algorithm which converts load sequences into curve functions. First, the load sequence with length *n* is divided into *m* segments with length *m*, where m=⌈n/k⌉. For the i-th segment of the load sequence with length *k*
$x_{k\times i},x_{k\times i+1},\ldots ,x_{(k+1)\times i-1}$, the starting sequence of the next segment is added into an augmented load sequence with length k+1$x_{k\times i},x_{k\times i+1},\ldots ,x_{(k+1)\times i}$. The last segment is processed separately. The *i*-th (i<m) segment of augmented load sequence is converted into *k* smooth cubic curves $f_{k\times i},f_{k\times i+1},\ldots ,f_{(k+1)\times i-1}$. They are represented as a piecewise function $S_{i}\left(t\right)$, as shown in (18):(18)$$S_{i}\left(t\right)=\left\{\begin{array}{c} f_{k\times i}\left(t\right),\;t_{k\times i}<t<t_{k\times i+1}\\ \cdots \\ f_{(k+1)\times i-1}\left(t\right),t_{(k+1)\times i-1}<t<t_{(k+1)\times i} \end{array}\right.$$

The complexity of the algorithm is $O(k^{2}n)$.

#### 3.2.3. The Similarity Measure of Load Sequence

For the similarity measure of load sequence, the original dynamic time bending distance has the following deficiencies: time insensitivity and high algorithm complexity. The similarity measure of the VM load is time-sensitive and isometric [24]. When solving for the smallest curved path, the algorithm can be optimized by limiting the search space of the smallest curved path so that the time complexity is O(wn), where *w* is the search width. The algorithm to measure similarity of load sequence is named dynamic function warping distance algorithm (DFWD).

For the function sequences $f_{1}^{A},f_{2}^{A},\ldots ,f_{n-1}^{A}$ and $f_{1}^{B},f_{2}^{B},\ldots ,f_{n-1}^{B}$ corresponding to the load sequences of two VMs *A* and *B*, we define the time-aware integral difference (TAID) of the function $f_{i}^{A}$ and function $f_{j}^{B}$, as shown in (19):(19)$$TAID_{ij}=\alpha^{\left|(j-i)\right|}\int_{0}^{T}\left|f_{i}^{A}\left(t\right)-f_{j}^{B}\left(t+\left(\left(j-i\right)\times T\right)\right)\right|dt$$
where *T* is the acquisition period of the load data, and α(α>1) is the distance sensitivity coefficient. The bigger α is, the more sensitive it is to the time when the load similarity measure is performed:(20)$$\begin{array}{c} \mathrm{DTW}\left(A,B\right)=\min_{W}\left\{\sum_{i=1}^{p}d\left(w_{i}\right),W=<w_{1},w_{2},\ldots w_{k}>\right\}\\ \;\mathrm{where}\;w_{i}=\left(a_{i},b_{i}\right)\\ d\left(w_{i}\right)=\left\{\begin{array}{c} TAID_{a_{i},b_{i}},\left|a_{i}-b_{i}\right|\leq \;\mathrm{SearchWidth}\;\\ +\infty ,\left|a_{i}-b_{i}\right|>\;\mathrm{SearchWidth}\; \end{array}\right. \end{array}$$

Equation (20) represents the path of the minimum dynamic function bending cost, that is, the best correspondence between load sequence values.

Where $w_{i}=\left(a_{i},b_{i}\right)$ is the i-th sequence value pair, and $d(w_{i})$ is the corresponding distance of $w_{i}=\left(a_{i},b_{i}\right)$. It is measured by the TAID, and the distance is infinite when the search width exceeds the threshold.

The DFWD between load sequences can be solved by dynamic programming. The cumulative distance matrix is shown in (21):(21)$$\begin{array}{c} r\left(i,j\right)=d_{w_{i}}+\min\left\{r\left(i-1,j\right),r\left(i,j-1\right),r\left(i-1,j-1\right)\right\}\\ \;\mathrm{where}\;|i-j|\leq \;\mathrm{SearchWidth} \end{array}$$

At the end of the dynamic programming, r(n−2,n−2) represents the dynamic function bending distance. The dynamic function bending distance is the best matching relationship between two load sequences. The greater the dynamic function bending distance, the smaller the similarity of the two load sequences. The DFWD algorithm is shown as Algorithm 2.
**Algorithm 2: **DFWD(lsA, lsB)**Input:** 
Load sequences lsA and lsB**Output:** 
Dynamic function bending distance distance1:Standardization of lsA and lsB2:Calculate conversion function3:r[n−1][n−1]4:**for**i=0 to n−2 **do**5:     **for** j=0 to n−2 **do**6:         **if** abs(i,j)<SearchWidth **then**7:            r[i][j]=Double.Max_Value8:         **else**9:            Calculate taid10:          Calculate r[i][j]=taid+min{r[i−1][j],r[i][j−1],r[i−1][j−1]}11:       **end if**12:    **end for**13:**end for**14:**return**r[n−2][n−2]

   To predict the load sequence of the VM, we proposed an algorithm called load similarity match predicted (LSMP) based on the proposed similarity measurement algorithm DFWD. The LSMP algorithm is shown as Algorithm 3.
**Algorithm 3: **LSMP(*v*, *k*)**Input:** 
VM *v*, Prediction length *k***Output:** 
Load sequence of *v* in the future *k* consolidation periods $ls_{future}$1:Initialize $ls_{history},minDistance$2:Get generated load sequence $ls_{before}\left[n\right]$3:Get past m-day load sequence $LS_{history}$4:**for**$ls\in LS_{history}$**do**5:    $distance=DFWD(ls_{before}\left[n\right],ls)$6:    **if** distance<minDistance **then**7:         minDistance=distance8:         $ls_{history}=ls$9:    **end if**10:**end for**11:Initialize $ls_{match},minDistance$12:Get latest *k* consolidation cycles load sequence $ls_{before}$13:**for**i=0 to w∗2 **do**14:    Calculate ls[k]15:    $distance=DFWD(ls_{before},ls\left[k\right])$16:    **if** distance<minDistance **then**17:         minDistance=distance18:         $ls_{match}=ls\left[k\right]$19:    **end if**20:**end for**21:Calculate $ls_{future}$22:**return** $ls_{future}$ = 0

Algorithm 3 first uses the CPU load sequence ls generated by the VM *v* today to find the historical load sequence $ls_{history}$ with the highest similarity in the same time period through the similarity measure. Then, it uses the load sequence $ls_{before}$ of the most recent *k* scheduling points to match the isometric load sequence $ls_{match}$ with the highest similarity. Finally, it uses the load sequence $ls_{future}$ in $ls_{match}$ of $ls_{history}$ in the subsequent *k* consolidation cycles to represent the CPU utilization for the next *k* consolidation cycles and adjusts the amplitude. The time complexity of Algorithm 3 is O(n), where *n* is the number of consolidation cycles during a day.

### 3.3. VMs to Be Migrated Selection Strategy

Program flow chart is shown in Figure 2.

The CPU load increment of VM is predicted by the VMTRP algorithm. After this process, the VMs set on the overloaded PM is divided into three parts: a load-stationary VMs set, a growth-stable VMs set, and a growth-significant VMs set. The priority of these sets increases in turn. We search from the VMs set with the highest priority. If there is a certain VM such that the PM predicted load fitness becomes positive after it is moved out, then the virtual machine is chosen to be migrated out, and the program ends. Otherwise, selecting the VMs with the highest CPU load in the set for migration, we repeat the previous step.

#### 3.3.1. Design of Load Fitness Function

The CPU load threshold of the overloaded PM is denoted by *H*. The load threshold is only set to prevent further increases in the PM load and reduce the risk of SLAV. If CPU load is close to the threshold, then the probability of overloading in the next cycle would be large. As a result, frequent VM migrations may occur, and the performance of applications will be affected.

Therefore, we use a fitness function to examine the PM load level, which is given in (22):(22)$$\begin{array}{c} f\left(u\right)=\left\{\begin{array}{cc} \frac{m}{g}u & ,0\leq u\leq g\\ \frac{m}{\log\frac{1-g}{1-H}}\log\frac{1-u}{1-H} & ,g<u<1\\ -1 & ,u=1 \end{array}\right. \end{array}$$

#### 3.3.2. PM Load Prediction

The set of all the VMs on the overloaded PM is denoted by $V\left\{v_{0},v_{2},\ldots ,v_{n-1}\right\}$. The set of VMs to be migrated is denoted by $V_{out}$, and the set of remaining VMs is denoted by $V_{rest}$. According to the (23), the total load of the VMs set for a specific consolidation cycle can be calculated:(23)$$\begin{array}{c} VL=\sum_{i=0}^{n-1}\sum_{j=0}^{m-1}u_{ij}\times T \end{array}$$
where $u_{ij}$ is CPU utilization of the *j*-th acquisition point at a specific cycle of the VM $v_{i}$.

The sets of VMs with a growing load in the VMs set *V* and in the VMs set to be migrated $V_{out}$ are respectively denoted by $V_{UP}$ and $V_{up}$. First, according to (18), the total load $VL_{now},VL_{pre},VL_{now}^{UP}\;and\;VL_{pre}^{UP}$ of the VMs set *V* and $V_{UP}$ in the current consolidation cycle and in the previous consolidation cycle, respectively, is calculated. Then, according to (24), the conversion ratio of the VMs set $V_{UP}$ load increment to the PM load increment is calculated:(24)$$cr=\frac{VL_{now}^{UP}-VL_{pre}^{UP}}{VL_{now}-VL_{pre}}$$

Then, the predicted load increments $LI_{UP}$ and $LI_{up}$ for the VM set $V_{UP}$ and $V_{up}$ are calculated, respectively. Finally, the predicted load of the remaining VMs set $V_{rest}\left(v_{0},v_{1},\ldots ,v_{m-1}\right)$ is calculated based on (25):(25)$$\begin{array}{c} u_{\mathrm{predict}}=\frac{cr\times LI_{UP}+T\times \sum_{i=0}^{m-1}u_{i}-LI_{up}}{T} \end{array}$$
where *m* represents the number of VMs in $V_{rest}$, $u_{i}$ indicates the current CPU utilization of the VM $v_{i}$, and $cr*LI_{UP}$ denotes the predicted load increments when the PM does not perform any VM migration. The time complexity is O(n∗m), where *n* is the number of sampling periods in the consolidation cycle, and *m* is the number of VMs in the overloaded PM.

The LIP strategy selects the VMs to be migrated from the above three sets in order according to the priority through a two-stage greedy selection. The LIP Algorithm is shown as Algorithm 4.

The time complexity of Algorithm 4 is $O(n*m^{3})$, where *n* is the number of acquisition periods in the consolidation period, and *m* is the number of VMs in the overloaded PM. Since the aggregate resource demands of VMs in the same PM do not exceed the resource capacity of the PM, the number of VMs deployed on the PM is limited [25]. Therefore, *m* is small, and the time complexity of this algorithm is acceptable.

The LIP algorithm improves the accuracy of the migrated VMs selection by combining the predicted load increments and current load. On the one hand, it can effectively prevent the massive resource fragmentation caused by the improper selection of migrated VMs. On the other hand, it can effectively reduce the probability of the PM being overloaded again and reduce the number of migrations.
**Algorithm 4: **LIP($V_{l},V_{m},V_{h}$)**Input:** 
The stable load VMs set $V_{l}$, The stable growth load VMs set $V_{m}$, The significant growth load VMs set $V_{h}$**Output:** 
The VMs set to be migrated $V_{out}$1:**while**$u_{predict}\left(V-V_{out}\right)$ is overload **do**2:    Get $V_{c}$ (with the highest priority in $V_{l},V_{m},V_{h}$)3:    Initialize $success,maxFit,v_{b}$4:    **for** $v\in V_{c}$ **do**5:        **if** $PMLP(V-V_{out}-v)$ is not overload **then**6:             Calculate fit7:        **end if**8:        **if** fit>maxFit **then**9:             maxFit=fit10:            $v_{b}=v$11:            success=true12:        **end if**13:        **if** success=true **then**14:            $V_{\mathrm{out}\;}\leftarrow V_{\mathrm{out}\;}⋃v_{b}$15:            $V_{c}\leftarrow V_{c}-v_{b}$16:            Return $V_{out}$17:        **else**18:            Get $V_{h}$ (with the highest priority in $V_{c}$)19:            $V_{\mathrm{out}\;}\leftarrow V_{\mathrm{out}\;}⋃v_{b}$20:            $V_{c}\leftarrow V_{c}-v_{b}$21:        **end if**22:     **end for**23:**end while**24:**return**$V_{out}$ =0

### 3.4. VMs Migration Destination Selection Strategy

After selecting the VMs to be migrated from the overloaded PM, we need to select a suitable target PM for each VM. For a specific VM $v_{m}$, and a set of target PMs $H(h_{1},h_{2},\ldots ,h_{p})$, the PM with the largest saturation increase rate after migration needs to be selected as the VM migration point. That is, it is necessary to select such a PM $h_{j}$ that meets (26):(26)$$\begin{array}{c} h_{j}\in H|\forall h_{a}\in H,{\mathrm{SIR}}_{a}\leq SIR_{j} \end{array}$$

The traditional VMC algorithm always uses a static constant to characterize the load of a VM. If the load level of the VM is characterized by the mean, although the resource utilization of the PM can be effectively improved, the probability of the loads peak overlap among the VMs increases [26].

The VMC algorithm with static parameters cannot sense the dynamic characteristics of the load sequence, so the complementary characteristics between the load sequences cannot be used for further resource optimization. Some algorithms have begun to consider characterizing VMs through resource load sequences. However, these schemes only perform complementary matching of resources before the consolidation of VMs, and there is still no complementary matching detection for the resource load sequence in the consolidation process.

Therefore, we proposed the SIR algorithm to select the best migration point by calculating the saturation increase ratio of the PM sequence before and after the VM migration. The strategy consists of three steps. First, predict the load sequence of the PM after migrating the VM to this PM. Then calculate the saturation increase ratio according to Equation (29). Finally, the PM with the largest saturation increase ratio is selected as the destination point. Next, we introduce the concepts of saturation and saturation increase ratio.

#### 3.4.1. Saturation

For the VMs set $V(v_{1},v_{2},\ldots ,v_{k})$, the load sequence composed of the past *k* and the future *k* scheduling points load sequence of VM $v_{i}$ is represented as $X_{i}\left(x_{i}\left(t_{1}\right),x_{i}\left(t_{2}\right),\ldots ,x_{i}\left(t_{2k}\right)\right)$. The load sequence of the VMs set *V* is $Y(y\left(t_{1}\right),y\left(t_{2}\right),\ldots ,y\left(t_{2k}\right))$, as shown in (27):(27)$$\begin{array}{c} Y\left(t_{j}\right)=\sum_{i=1}^{2k}x_{i}\left(t_{j}\right),1<j<m \end{array}$$

The load saturation of the VMs set *V* is shown in (28):(28)$$\begin{array}{c} S=\frac{\mu }{\mu +2\sigma } \end{array}$$
where μ is the mean of the load data set $\left\{y\left(t_{1}\right),y\left(t_{2}\right),\ldots ,y\left(t_{2k}\right)\right\}$, and $\sigma^{2}$ is the variance.

#### 3.4.2. Saturation Increase Rate

The VMs set on the PM $H_{j}$ is denoted by $V(v_{1},v_{2},\ldots ,v_{k})$. The load mean value and saturation of the VMs set *V* are denoted by $\mu_{pre}$ and $S_{pre}$. The load mean value and saturation of the VMs set $V^{\prime }=V\cup v$ after the VM *v* is migrated to the PM $H_{j}$ are denoted by $\mu_{post}$ and $S_{post}$. Then the saturation increase rate is shown as (29):(29)$$\begin{array}{c} SIR_{j}=\left(s_{post}-s_{pre}\right)\times \frac{u_{pre}}{u_{post}-u_{pre}} \end{array}$$

The saturation increase rate calculated by (29) can effectively shield the difference in the PM load level. The higher the PM load, the weaker the saturation increase.

The SIR algorithm is shown as Algorithm 5.
**Algorithm 5: **SIR($v_{m},H$)**Input:** 
VM to be migrated $v_{m}$, Target PMs set $H(h_{1},h_{2},\ldots ,h_{p})$**Output:** 
VM migration point $h_{j}$1:Initialize $h_{j},maxSIR$2:$lsm\left[2k\right]=LSMP\left(v_{m}\right)$3:**for**h∈H**do**4:    **for** v∈V **do**5:        **for** i=0 to 2k−1 **do**6:            LS[i]+=ls[i]7:        **end for**8:     **end for**9:     Calculate sPre,sPost10:   Calculate sir11:   **if** sir>maxSIR **then**12:       $h_{j}\leftarrow h$13:       maxSIR=sir14:   **end if**15:**end for**15:**return**$h_{j}$

First, we use the LSMP algorithm to predict the load sequence of the future *k* scheduling points of the VM $v_{m}$. Then the load sequence lsm with length 2k is obtained by combining the load sequence of the past *k* scheduling points. For any PM *h* in the PMs set *H*, the overall load sequence LS of the PM is obtained by accumulating the load sequences of the VMs. Then the saturation increase rate is calculated. The time complexity of Algorithm 5 is O(n∗m), where *n* represents the number of VMs to be migrated and *m* represents the length of the predicted sequence.

### 3.5. Virtual Machines Consolidation

Our VMC algorithm based on load prediction contains the following two parts.

#### 3.5.1. Over-Loaded Host Process

The goal of this process is to migrate VMs on over-loaded PMs to reduce SLAV. For the active PM set *H*, the overloaded PM set $H_{0}$ is first obtained through comparing the current load to threshold, and the remaining PM set is called $H_{rest}$. We use the LIP strategy to select the set of VMs $V_{m}$ to be migrated for each PM $h_{s}$ in $H_{0}$. All VMs are ranked in descending order of CPU load. For each VM in $V_{m}$, the best migration point $h_{best}$ is selected, then $<h_{s},v,h_{best}>$ is added to the migration plan *M* and the resource information of the PM $h_{d}$ is updated. If a suitable VM migration point cannot be found, a new PM $h_{new}$ will start. $<h_{s},v,h_{new}>$ will be added to the migration plan *M*, and $h_{new}$ will be added to *H*. The algorithm is shown as Algorithm 6.
**Algorithm 6: **overUtilizedHostProcess(*H*)**Input:** 
Active PM set *H***Output:** 
Migration plan *M*1:Calculate $H_{0}$2:$H_{rest}=H-H_{0}$3:**for**$h_{s}\in H_{0}$**do**4:     Calculate $V_{m}$ by LIP5:     **for** $v\in V_{m}$ **do**6:         maxSIR=Double.MIN7:         maxHost=NULL8:         **for** $h_{d}\in H$ **do**9:              **if** $v\left(c,r,b\right)+\left(c,r,b\right)<Thr\left(c,r,b,h_{d}\right)$ **then**10:                 Calculate sir11:                 **if** sir>maxSIR **then**12:                     maxSIR=sir13:                     $maxHost=h_{d}$14:                 **end if**15:             **end if**16:        **end for**17:        **if** maxHost!=NULL **then**18:             $M=M⋃<h_{s},v,maxHost>$19:             Update(maxHost(c,r,b))20:        **else**21:             newHost22:             $M=M⋃<h_{s},v,newHost>$23:             Update(newHost(c,r,b))24:         **end if**25:     **end for**26:**end for**27:**return***M*

The time complexity of Algorithm 6 is O(n∗m∗k), where *n* represents the number of VMs to be migrated, *m* represents the number of target PMs, and *k* represents the length of the predicted sequence.

#### 3.5.2. Under-Loaded Host Process

The goal of this process is to migrate out all VMs on low-load PMs to reduce energy consumption. First, the set of non-overload PMs $H_{rest}$ obtained from the over-utilized host process is ranked in descending load. The algorithm starts from the PM with the lowest load. We use the SIR strategy to select the appropriate migration point $h_{d}$ for each VM and update the PM resources after a successful selection. If a VM cannot find a suitable migration point, all previous attempts need to be undone, and the algorithm will be terminated. If suitable migration points can be found for all the VMs on the PM $h_{s}$, they will be migrated out, and the resources update of the PM $h_{s}$ during the trial will be saved. At the same time, $<h_{s},v,h_{d}>$ will be added to the migration plan, and the PM will be shut down to save energy. The algorithm is showed as Algorithm 7.

The time complexity of Algorithm 7 is O(n∗m∗k∗p), where *n* represents the number of under load PMs, *m* represents the number of target PMs, *k* represents the length of the predicted sequence, and *p* represents the number of VMs on PM.
**Algorithm 7: **underUtilizedHostProcess($H_{rest},M$)**Input:** 
Non-over load PMs set $H_{rest}$, migration plan *M***Output:** 
Migration plan *M*1:Sort $H_{rest}$2:**for**$h_{s}\in H_{rest}$**do**3:    **if** $h_{s}>H$ **then**4:        Return *M*5:    **end if**6:    success=true7:    Sort VM8:    **for** v∈VM **do**9:         Calculate $h_{d}$ by SIR10:        **if** $h_{d}==NULL$ **then**11:            success=false;break12:        **end if**13:        $M=M⋃\{<h_{s},v,h_{d}>\}$14:        Update($h_{d}\left(c,r,b\right)$)15:        **if** success==true **then**16:            Shutdown($h_{d}$)17:        **else**18:            Recover($h_{s}$)19:            Return *M*20:        **end if**21:     **end for**22:**end for**23:**return***M*

## 4. Performance Analysis

### 4.1. Experiment Environment

CloudSim was used as a simulation platform that has 800 heterogeneous PMs and supports 800 VMs running on it. The simulation experiments lasted 24 h, and the consolidation period was 300 s. Two host configurations were selected: HP G4 (VM monitor: Xen, processor: 2 × 1860 MHz, memory: 4 GB, network bandwidth: 1 GB/s) and HP G5 (VM monitor: Xen, processor: 2 × 2660 MHz, memory: 4 GB, network bandwidth: 1 GB/s). In the experiments, the number of G4 servers and G5 servers each accounted for half.

The simulation experiments used four types of VMs provided by AmazonEC2, as shown in Table 3. The real load data used in the simulation experiment are derived from single-core VMs. Therefore, the VMs in the experiment are all set to single-core. Specifically, each VM is allocated 30% of its requested CPU resource when it is initialized. Next, in the consolidation cycle, resources are allocated to each VM according to the resource request of the load history.

To ensure the validity of the simulation experiment evaluation results, we used real data provided by the CoMon project [27], which contains the real CPU load data of over a thousand VMs from more than 500 locations around the world. The load data are obtained by collecting every five minutes. In the simulation experiment, the real operating environment of the data center is reproduced by binding the real resource load mode (Table 4).

### 4.2. Evaluation Index

The goal of the VMC algorithm includes the following: (1) Ensure that SLA is not violated. (2) Minimize energy consumption. (3) Minimize the number of VM migrations. Therefore, the performance of the algorithm is evaluated by the following indicators: SLAV, EC, NOM, COM and EPB.

#### 4.2.1. SLAV (SLA Violation)

Beloglazov et al. [28] proposed a load-independent SLA violation evaluation standard. Here, the SLA violations due to over-utilization caused by competition for PM resources are focused. Therefore, SLAV is defined as the proportion of time that the CPU is fully loaded, as shown in (30):(30)$$\begin{array}{c} SLAV=\frac{\sum_{i=1}^{M}T_{s_{i}}}{\sum_{i=1}^{M}T_{a_{i}}} \end{array}$$
where *M* is the number of PMs, $T_{s_{i}}$ is the total time that the CPU is fully loaded for the PM *i*, and $T_{a_{i}}$ is the total time that the PM *i* is in an active state.

#### 4.2.2. EC (Energy Consumption)

In this paper, the overall energy consumption of the data center EC is the total energy consumption of each PM. The energy consumption of a PM is calculated by its CPU utilization [29], as shown in (31) and (32):(31)$$\begin{array}{c} EC=\sum_{i=0}^{M}E_{i} \end{array}$$
(32)$$\begin{array}{c} E_{i}=\sum_{j=1}^{n}U_{i}^{c}\left(t_{j}\right)\times T\times P\left(U_{i}^{c}\left(t_{j}\right)\right) \end{array}$$
where $E_{i}$ is the energy consumption of PM *i*, *T* is the acquisition period for the PM load, $U_{i}^{c}\left(t_{j}\right)$ is the CPU utilization of the PM *i* at the moment $t_{j}$, and $P\left(U_{i}^{c}\left(t_{j}\right)\right)$ is the PM power corresponding to CPU utilization.

The energy consumption models of PM G4 and G5 in the simulation experiments made the experimental results more credible by using the real energy consumption data provided by the SPECpower benchmark.

#### 4.2.3. NOM (Number of Migration)

NOM is the number of migrations. The migration process of VMs will bring additional overhead to the system, including the CPU resources of the source PM, the bandwidth between the source PM and the destination PM, and the service downtime in the process. Therefore, the value of NOM should be as small as possible.

#### 4.2.4. COM (Cost of Migration)

Most current studies only measure the quality of the algorithm by the NOM. Experiments in [9] show that the performance degradation and pause time caused by virtual machine migration depends on the memory size and CPU usage of the virtual machine. The larger the virtual memory, the longer the migration time required for dynamic virtual machine migration under certain network bandwidth conditions. The higher the CPU usage of the virtual machine, the higher the memory update frequency, and the more dirty pages generated during the memory migration process, which not only increases the transfer time and system overhead, but also increases the pause time. This paper used COM to evaluate the efficiency of VM selection algorithms and VM migration points selection algorithms. The COM of VM *i* is shown in (33) and (34):(33)$$\begin{array}{c} COM_{i}=T_{m_{i}}\times \left(1+k\times U_{i}^{c}\right)\\ T_{m_{i}}=\frac{M_{i}}{B_{i}} \end{array}$$
where $T_{m_{i}}$ is the time required for the migration of the VM *i* under normal circumstances, $B_{i}$ is the network bandwidth of the PM, $M_{i}$ is the memory size of the VM *i*, $U_{i}^{c}$ is the CPU usage of the VM *i*, and the parameter *k* is the CPU load influence factor that is the extra overhead for the VM migration caused by the VM maintaining the execution state in the VM migration process. The experiment is set as k=1.

#### 4.2.5. EPB (Energy Performance Balance)

The goal of VMC technology is to achieve the best balance between energy consumption and service performance. The main metrics are energy consumption, SLAV and cost of migration; however, these metrics are typically negatively correlated. Therefore, we propose a combined metric that captures all these three metrics. The EPB can comprehensively evaluate the algorithm, as shown in (35):(34)$$\begin{array}{c} EPB=EC\times SLAV\times COM \end{array}$$

### 4.3. The Results Analysis

#### 4.3.1. The Evaluation of Migration VM Selection Strategy

Our LIP algorithm was compared with random selection (RS) [9], maximum load (Max), minimum load (Min), and minimum migration time (MMT) [9]. In order to exclude the influence of the VM migration point selection strategy, the random selection strategy was adopted uniformly in this experiment.

(a)EC (energy consumption) comparison

Figure 3a shows the energy consumption of different migration VM selection strategies under different PM load thresholds. Compared to Max, RS, Min, and MMT, LIP reduced energy consumption by approximately 10%, 25%, 45%, and 30%, respectively. It can be seen that the overall energy consumption of the data center presents a downward trend as the load threshold increases. In addition, choosing VM with a relatively high load to migrate causes less energy consumption. The reasons are due to two aspects. First, when the load of the VM is small, it is easy to use the remaining fragment resources to select the target PM. Therefore, it is easier to minimize the number of PMs in the process of VMC and reduce energy consumption. However, VM load increasing is more likely to happen if the VM load is too small. The load of the data center will be unbalanced due to the dynamic changes of the load after the integration. Finally, it will reduce the average resource utilization and increase energy consumption of the data center. Second, the more frequent VMC causes the PM state to be switched more frequently, which needs to consume additional energy [30,31].

(b)SLAV (SLA violation) comparison

Figure 3b shows the SLAV under different PM load thresholds for different migration VM selection strategies. Compared to Max, RS, Min, and MMT, LIP reduced SLAV by approximately 55%, 60%, 59%, and 53%, respectively. The SLAV of LIP is significantly smaller than those of the other algorithms. In particular, when the load threshold is close to 100%, the SLAV of LIP does not drastically increase. This is because LIP has an overall analysis and prediction of the VM load change trend, which kept the PM load as saturated as possible after the VM migration, thereby effectively reducing the probability of the PM resource competition.

(c)NOM (Number of Migration) and COM (Cost of Migration) Comparison

Figure 3c,d show the number of VM migrations (NOM) and the cost of migration (COM) for different migration VM selection strategies under different PM load thresholds. Compared to Max, RS, Min, and MMT, LIP reduced the cost of migration by approximately 43%, 61%, 66%, and 57%, respectively.

The experimental result shows that when the VM with high CPU load is selected to be migrated, the number of migration and cost of migration are relatively low. The reason is that it significantly eases the competition for the CPU resources on the PM. Therefore, the number of VMs to be migrated will be effectively reduced. In addition, MMT differs drastically from RS in terms of NOM and COM. Because the overall load of VMs with low COM is relatively small, frequent VMC is likely to occur, resulting in more VM migrations. However, due to the short migration time, the cost is effectively reduced.

(d)EPB (energy performance balance) comparison

Figure 3e shows EPB for each migration VM selection strategy under different PM load thresholds. Compared to Max, RS, Min, and MMT, the EPB of LIP was reduced by approximately 75%, 88%, 91%, and 87%, respectively. As can be seen from Figure 3e, energy consumption and service performance reach the best balance when the load threshold is around 0.8. In the process of increasing the load threshold, the EPB of LIP remains stable, which fully demonstrates the accuracy of LIP in the selection of the migration VM.

#### 4.3.2. The Evaluation of VM Migration Point Selection Strategy

Our SIR algorithm was compared with random selection (RS) [9], minimum energy increase (MEI), and minimum resource fragmentation (MRF). In order to exclude the influence of the migration VM selection strategy, the random selection strategy was adopted uniformly in this experiment.

(a)EC (energy consumption) comparison

Figure 4a shows the energy consumption under different PM load thresholds for different strategies. Compared to MEI, RS, and MRF, SIR reduced energy consumption by approximately 30%, 41%, and 6%, respectively. Although the goal of MEI is to minimize the increase in energy, the energy consumption is higher than that of MRF and SIR. The reason is that MEI fails to fully utilize the resource fragmentation of the PM, resulting in a low utilization rate of the data center. The goal of MRF is to minimize resource fragmentation and has a significant reduction in energy consumption compared to RS and MRF. The goal of SIR is to increase the smoothness of the PM load sequence. When the PM load sequence is smooth, the utilization rate of PM will increase, therefore reducing the energy consumption of data center.

(b)SLAV (SLA violation) comparison

Figure 4b shows the SLAV of different strategies under different PM load thresholds. Compared to MEI, RS, and MRF, the SLAV of SIR was reduced by approximately 55%, 79%, and 64%, respectively. First, the SLAV increases with the increase of the load threshold. When the load threshold exceeds 0.8, the SLAV increases sharply. Secondly, the SLAV of MEI is lower than MRF. This is because MRF makes the PM resources fragmentation smaller, which increases the risk of resource competition. SIR makes the PM load more stable and effectively reduces the SLAV caused by the full load of the PM CPU.

(c)NOM (number of migration) and COM (cost of migration) comparison

Figure 4c,d respectively show NOM and COM for different strategies under different PM load thresholds. Compared with MEI, RS, and MRF, SIR reduced COM by approximately 23%, 87%, and 60%, respectively. SIR makes the PM load sequence smooth, and the performance improvement effect in terms of NOM and COM is obvious. When the load threshold reaches 0.9, NOM and COM of MEI dramatically increase because MEI fails to consider the intense competition of PM resources. Secondly, there is a clear consistency between NOM and COM because the choice of VM migration points does not affect the VM migration cost.

(d)EPB (energy performance balance) comparison

Figure 4e shows the EPB for different strategies under different PM load thresholds. Compared to MEI, RS, and MRF, SIR reduced energy consumption by approximately 91%, 77%, and 88%, respectively. It can be seen that energy consumption and service performance reach the optimal balance when the load threshold is around 0.8. SIR demonstrated performance improvements over the other strategies.

All the above experimental results show that the proposed VMC algorithm has better efficiency good in terms of accuracy, stability, and energy efficiency than existing schemes. Therefore, it can reduce the cloud computing pressure brought by the growth of IoT devices.

### 4.4. Engineering Applications

The cloud data center in the blockchain is composed of a large number of physical machines (PM), as shown in Figure 5. In order to provide diversified services, the cloud data center uses virtualization technology to construct a virtual resource pool of computing resources (CPU, memory, GPU, and FPGA), storage resources, and network resources (routing, and bandwidth).

Virtual machine integration technology is a good solution to the problem of high energy consumption in the blockchain. The idea is to migrate virtual machines on some physical machines to other active physical machines so that some physical machines switch to low-energy mode or sleep mode and finally reduce energy consumption in the blockchain. The VMC algorithm proposed in this paper can be widely used in existing cloud data processing centers used in blockchain. These cloud data center energy-saving methods based on virtual machine online migration and load-aware integration technologies can effectively reduce the number of physical servers actually required by the cloud data center, shut down physical servers running without load, increase the overall utilization of server resources, and achieve green energy saving.

## 5. Conclusions

With the development of blockchain and explosive growth of the number of IoT devices, with the existing real-time cloud computing, it is difficult to meet the current requirements in terms of accuracy, stability, and energy consumption [32]. This article creatively uses the predicted future load data of VMs and PMs for virtual machine consolidation, improving the performance of mobile cloud computing. The experimental results showed that the performance of LIP migration virtual machine selection strategy and SIR virtual machine migration point selection strategy can be significantly improved over other strategies in terms of various evaluation indexes. This means that the proposed VMC algorithm can effectively improve the overall service performance of cloud computing in blockchain, including good accuracy, good stability, and energy efficiency.

Since the resources of virtual machines include not only the CPU load, but also the memory and network bandwidth, their competition for hardware resources will also cause service level agreement violation (SLAV). Therefore, we need to further study the impact of the differences and the correlation between the load sequences of different resources on the efficiency of virtual machine consolidation. As IoT devices are being upgraded, we need to consider other factors that may affect the process of VMC to improve the universality of our method. The time complexity of the algorithm needs to be optimized since it is significantly related to the number of VMs and PMs. In addition, the accuracy of the load prediction algorithm needs to be further improved based on real data from other projects.

## Figures and Tables

**Figure 1 entropy-25-00351-f001:**
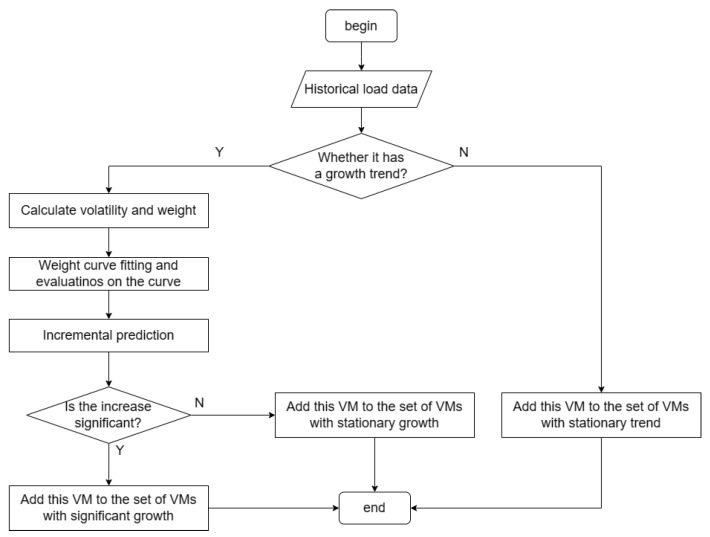
The flow chart of VWTRP Algorithm.

**Figure 2 entropy-25-00351-f002:**
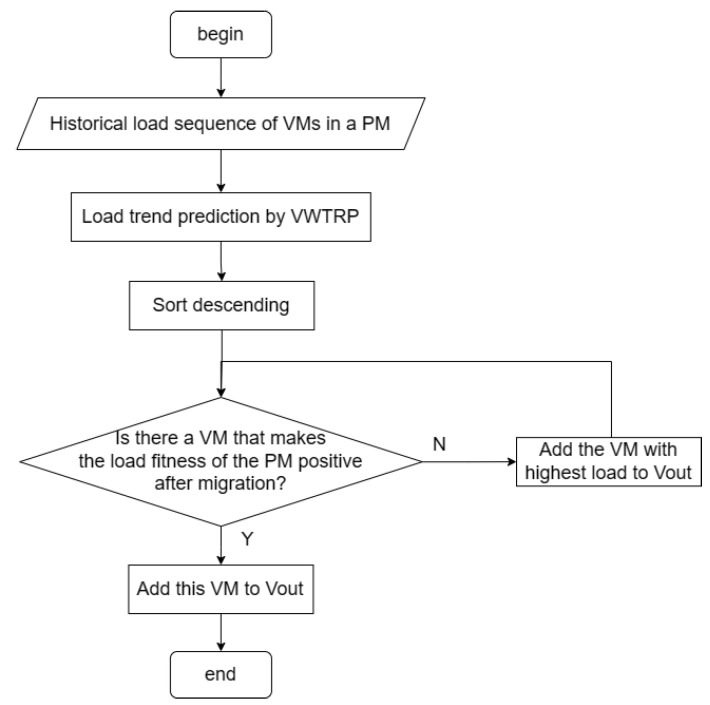
The flow chart of VMs to be migrated selection strategy.

**Figure 3 entropy-25-00351-f003:**
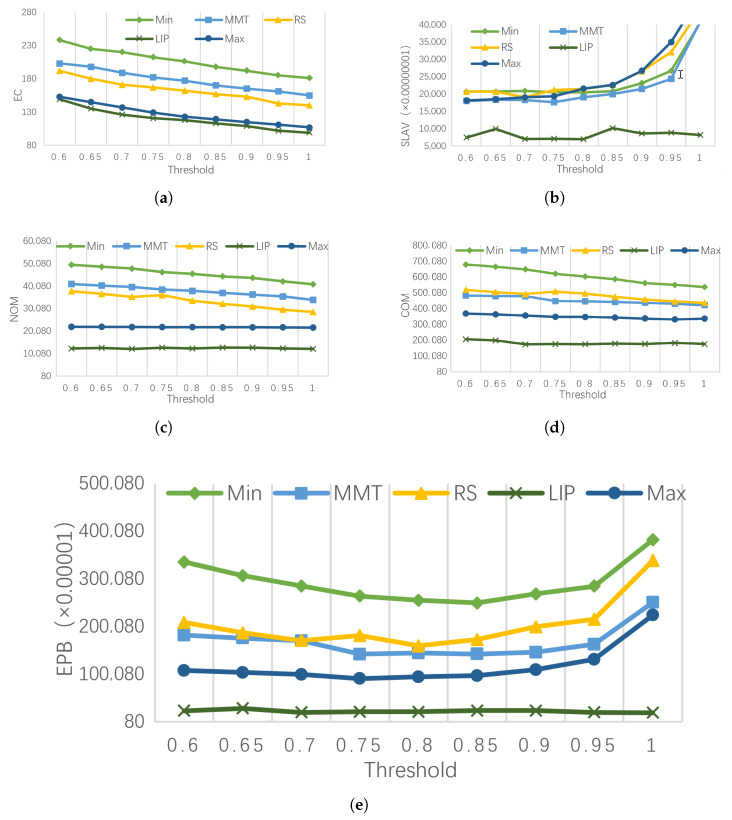
Performance comparison of LIP algorithms under different indicators. (**a**) The EC of different migration VM selection strategies. (**b**) The SLAV of different migration VM selection strategies. (**c**) The NOM of different migration VM selection strategies. (**d**) The COM of different migration VM selection strategies. (**e**) The EPB of different migration VM selection strategies.

**Figure 4 entropy-25-00351-f004:**
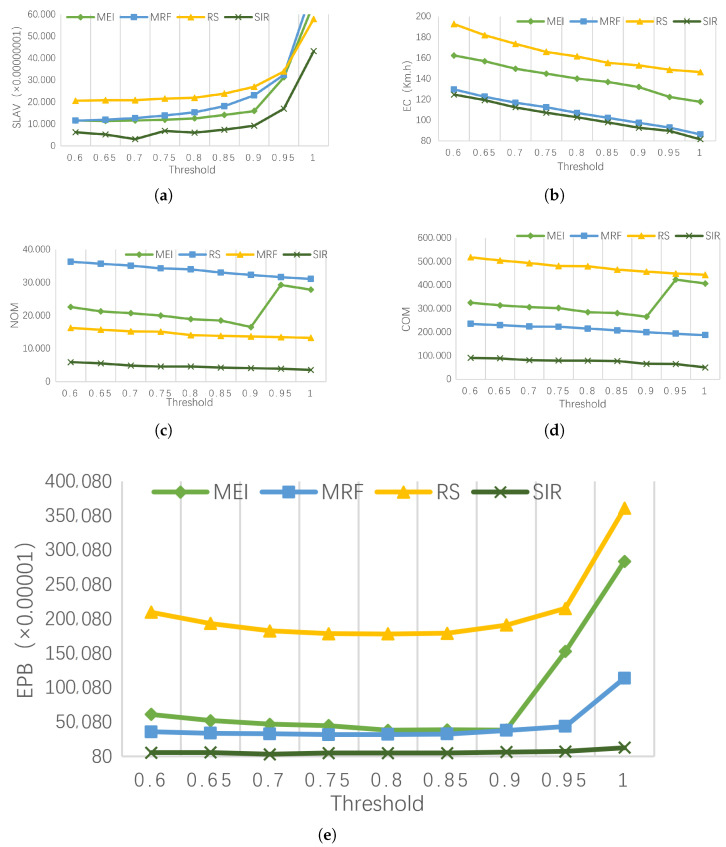
Performance comparison of SIR algorithms under different indicators. (**a**) The EC of different VM migration point selection strategies. (**b**) The SLAV of different VM migration point selection strategies. (**c**) The NOM of different VM migration point selection strategies. (**d**) The COM of different VM migration point selection strategies. (**e**) The EPB of different VM migration point selection strategies.

**Figure 5 entropy-25-00351-f005:**
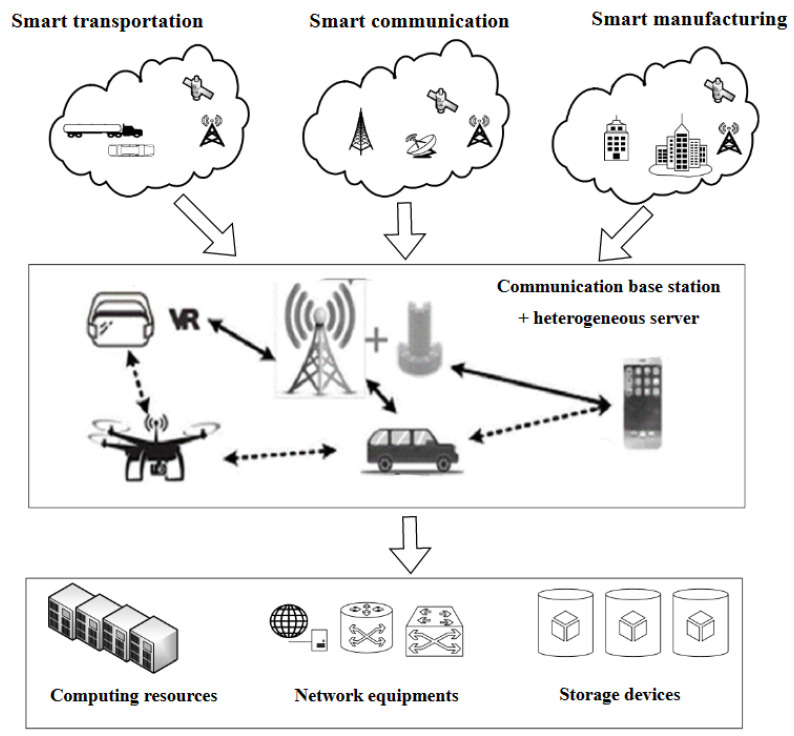
Application of vmc algorithm in industrial cloud computing center and blockchain.

**Table 1 entropy-25-00351-t001:** Abbreviations of proposed algorithms.

Abbreviation	Full Name and Description
VWTRP	Volatility-based Weighted Time Regressive Prediction. To predict short-term load trend for a VM
DFWD	Dynamic Function Warping Distance. To measure similarity of load sequence for a VM
LSMP	Load Similarity Match Predicted. To predict the load sequence of a VM
LIP	Load Increment Prediction. To select VMs that need to be migrated out
SIR	Saturation Increase Rate. To select the migration point for VMs

**Table 2 entropy-25-00351-t002:** The table of notations.

Notation	Meaning
*v*	VM
LH	Historical load data of *v*
LI	Predicted load increment of *v*
ls[n+1]	Load sequence
*A*	The number of reverse orders
*Z*	A statistic defined in Equation (3)
v[i]	The volatility of acquisition point *i*
w[i]	The weight of acquisition point *i*
curve(t)	Fitted curve
$R^{w}$	The goodness of fitting defined in Equation (10)
$V_{l}$	The load-stationary VMs set
$V_{m}$	The growth-stable VMs set
$V_{h}$	The growth-significant VMs set
distance	Dynamic function bending distance
$ls_{before}\left[n\right]$	Load sequence today $ls_{before}\left[n\right]$
$LS_{history}$	Load sequence in the past m days $LS_{history}$
$ls_{history}$	The most similar sequence of $ls_{before}\left[n\right]$ in $LS_{history}$
$ls_{before}$	Load sequence in latest *k* consolidation cycles
ls[k]	Load sequence after translation of $ls_{history}$ for i-w integration cycles in time period corresponding to $ls_{before}\left[n\right]$
$ls_{match}$	The most similar sequence of $ls_{before}$ in ls[k]
$ls_{future}$	Predicted load sequence
$V_{out}$	The VMs set to be migrated out
$u_{predict}\left(V-V_{out}\right)$	Predicted load of PM after $V_{out}$ migrated out
LS	Load sequence of PM
sir	Saturation increase rate defined in Equation (29)
$h_{j}$	VM migration point

**Table 3 entropy-25-00351-t003:** Configuration information of VMs.

	High-CPU	Large	Small	Micro
Kernel	1	1	1	1
RAM (MB)	870	1740	1740	613
Disk (GB)	2.5	2.5	2.5	2.5
Bandwidth (MB/s)	100	100	100	100
Frequency (MIPS)	2500	2000	1000	500

**Table 4 entropy-25-00351-t004:** The characteristics of load data.

Date	VMs Number	Mean	Standard Deviation	Median
3 March 2011	1052	12.31%	17.09%	6%
6 March 2011	898	11.44%	16.83%	5%
9 March 2011	1061	10.70%	15.57%	4%
22 March 2011	1516	9.26%	12.78%	5%
25 March 2011	1078	10.56%	14.14%	6%
3 April 2011	1463	12.39%	16.55%	6%
9 April 2011	1358	11.12%	15.09%	6%
11 April 2011	1233	11.56%	15.07%	6%
12 April 2011	1054	11.54%	15.15%	6%
20 April 2011	1033	10.43%	15.21%	4%

## Data Availability

The real load data provided by the CoMon project [27] come from the real CPU load data of over a thousand VMs in more than 500 locations around the world.

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
