# Peer review of "An Efficient Virtual Machine Consolidation Algorithm for Cloud Computing"

_entropy, 2023, doi:10.3390/e25020351_

Round 1
Reviewer 1 Report
The abbreviations of the proposed algorithms should be clearly described in the abstract and introduction and throughout the manuscript.
The related works description is not reasonable, more declarations are required to support the authors claims.
Better resolution for Figures 1,2, and 5 is needed.
Some assumptions require more clarifications. For example “the curve fitting should have a great weight”
References should be added to support the assumptions. For example “The similarity measure of VM load is time-sensitive and isometric”
More precise explanation is required to state that “Due to the limited number of VMs deployed on the PM, m is small and the time complexity of this algorithm is acceptable”
Reviewer 2 Report
The paper presents a new approach to predicting future load data of VMs and PMs for virtual machine consolidation, improving the performance of mobile cloud computing. The method is based on the combination of the virtual machine's predicted load sequence and historical load sequence and utilizes a strategy named SIR to predict the load prediction.
The paper is well organized, and pseudo-codes with flow charts enrich the text and improve readability.
The experimental results are significant and show that the method is adapted to obtain good accuracy, stability, and energy efficiency.
Moderate English changes are required.
I suggest detailing the future development.
